

# Simulation of hearing loss can induce pitch shifts for complex tones

Issei Ichimiya and Hiroko Ichimiya

Ichimiya Clinic, Kitsuki City, Oita, Japan

## ABSTRACT

**Background**. Most studies on pitch shift provoked by hearing loss have been conducted using pure tones. However, many sounds encountered in everyday life are harmonic complex tones. In the present study, psychoacoustic experiments using complex tones were performed on healthy participants, and the possible mechanisms that cause pitch shift due to hearing loss are discussed.

**Methods**. Two experiments were performed in this study. In experiment 1, two tones were presented, and the participants were asked to select the tone that was higher in pitch. Partials with frequencies less than 250, 500, 750, or 1,000 Hz were eliminated from the harmonic complex tones and used as test tones to simulate low-tone hearing loss. Each tone pair was constructed such that the tone with a lower fundamental frequency (F0) was higher in terms of the frequency of the lowest partial. Furthermore, partials whose frequencies were greater than 1,300 or 1,600 Hz were also eliminated from these test tones to simulate high-tone hearing loss or modified sounds that patients may hear in everyday life. When a tone with a lower F0 was perceived as higher in pitch, it was considered a pitch shift from the expected tone. In experiment 2, tonal sequences were constructed to create a passage of the song "Lightly Row." Similar to experiment 1, partials of harmonic complex tones were eliminated from the tones. After listening to these tonal sequences, the participants were asked if the sequences sounded correct based on the melody or off-key.

**Results**. The results showed that pitch shifts and the melody sound off-key when lower partials are eliminated from complex tones, especially when a greater number of high-frequency components are eliminated.

**Conclusion**. Considering that these experiments were performed on healthy participants, the results suggest that the pitch shifts from the expected tone when patients with hearing loss hear certain complex tones, regardless of the underlying etiology of the hearing loss.

## INTRODUCTION

Patients with hearing loss may perceive pitch shift when they hear tones; however, this pitch shift does not appear to be common knowledge among otologists and audiologists. *Sacks (2007)* discussed the case of an older male patient, who was a music composer with high-tone hearing loss and pitch shift. The patient noticed that the upper register of his piano was grossly out of tune. One of his concerns was that he, as well as any of

Corresponding author
Issei Ichimiya, ich-oit@umin.ac.jp

the otologists or audiologists he consulted, had never met or heard of anyone else with a similar condition. Subsequently, they came across an article by another musician who had experienced similar symptoms and realized that such changes might go unnoticed by non-musicians and that professional musicians may be reluctant to mention experiencing such symptoms for fear of losing their standing in the field. Thus, they suspected that this condition was underreported.

Similar cases have been occasionally reported through general or website articles written by doctors or patients themselves. However, only a few scientific papers have detailed the cases of patients with pitch shift related to hearing loss. The lack of studies on pitch shift may be attributed to the difficulty in appropriately evaluating this phenomenon. Some studies have indicated pitch shifts in cases of unilateral hearing loss by using the opposite ear as a reference (*Albers & Wilson, 1968*; *Ogura et al., 2003*; *Brännström & Grenner, 2008*). However, no quantitative studies have been conducted in the case of bilateral hearing loss, possibly due to the lack of a reference ear, which makes evaluation more difficult.

This study aimed to show that the evaluation of pitch shift by hearing loss can be even more complicated when the types of stimulation tones are considered. Although most studies on pitch shift have been conducted using pure tones (*Albers & Wilson, 1968*; *Ogura et al., 2003*; *Brännström & Grenner, 2008*), many sounds encountered in everyday life, such as those produced by musical instruments and speech, are harmonic complex tones (*Moore, 2012*). A harmonic complex tone is composed of several pure tones, each of which has a frequency that is an integer multiple of the frequency of a common fundamental component. Pitch perception corresponds to the frequency for a pure tone but to the fundamental frequency (F0) for a harmonic complex tone. Interestingly, when a complex tone that physically lacks an F0 component is presented, some listeners perceive the pitch of the tone as F0. This is known as the missing fundamental phenomenon (*Zatorre, 1988*; *Kurylo et al., 1993*; *Galbraith, 1994*; *Paquette, Bourassa & Peretz, 1996*; *Schneider et al., 2005*; *Yost, 2009*; *Ladd et al., 2013*), which has also been referred to as "residue pitch" (*Schouten, 1940*), "periodicity pitch" (*Licklider, 1951*), or "virtual pitch" (*Terhardt, 1979*) in the literature. In contrast, other listeners perceive the pitch of the complex tone that lacks an F0 component based on the frequency of the partials that make up the tones (*Schneider et al., 2005*). For example, consider tone A, a tone complex of 800 and 1,000 Hz, and tone B, a tone complex of 750 and 1,000 Hz. Tone A consists of two partials that could be the 4th and 5th harmonic of 200 Hz, whereas tone B consists of two partials that could be the 3rd and 4th harmonic of 250 Hz. When tone A is presented followed by tone B, some listeners will report that the pitch sequence of the missing fundamental rises from 200 to 250 Hz, whereas other listeners will report one partial fall in pitch sequence (*Smoorenburg, 1970*). The former and latter groups of listeners are known as "synthetic" and "analytic" pitch listeners in the literature (*Houtsma & Fleuren, 1991*); however, in this article, we use the terms F0 and spectral responses, respectively, for the individual responses with reference to the study by *Ladd et al. (2013)*.

We hypothesized that such complicated characteristics in pitch perception may be related to the pitch shifts noticed by patients with hearing loss. Therefore, we simulated hearing loss in participants with normal hearing in the present study and investigated the

condition in which pitches shift from the expected tones on hearing complex tones. In the first part of the study (experiment 1), we examined pitch shifts by having the participants compare several pairs of tones, whereas in the second part of the study (experiment 2), participants were exposed to a melody that might have sounded off-key due to altered pitches to illustrate how such pitch shifts might impact everyday life.

## MATERIALS & METHODS

### Participants

A total of 43 participants, comprising 27 men and 16 women aged 19–60 years (mean $\pm$ standard deviation [SD] = 40.1 $\pm$ 12.9), were included in this study. Among them, five participated in experiment 1 only, four participated in experiment 2 only, and the remaining participated in both experiments. To examine the effect of age on the study outcomes, the participants were divided into two groups according to their age: those who were $\leq$ 39 years (mean $\pm$ SD = 28.4 $\pm$ 5.7) and those who were $\geq$ 40 years (mean $\pm$ SD = 51.2 $\pm$ 5.9). For experiment 1, 19 participants were included in the younger group, and 20 participants were included in the older group. For experiment 2, 18 participants were included in the younger group, and 20 participants were included in the older group. All participants had normal hearing with no confirmed neurological conditions and confirmed that they were able to hear all the tones used in the experiments. The study protocols were reviewed and approved by the Clinical Research Ethics Committee of Ichimiya Clinic (approval number: H3001-1). This study was conducted in accordance with the Declaration of Helsinki. Written informed consent was obtained from all participants prior to their inclusion in the study.

### Tone preparation and equipment

The stimulation tones were created using the publicly available software Wave Editor TWE (Yamaha Corporation, Tokyo, Japan). All tones were saved as WAV files (16 bits/44.1 kHz).

Eight tone pairs, which are shown in Fig. 1, were prepared for experiment 1. Each block in the columns in Fig. 1 represents a partial of the complex tones. The duration of each tone was set at 500 ms, with a 10-ms rise-and-fall time. These were the tones from which the higher and lower partials were eliminated from the harmonic complex tones. The F0 of each tone pair was fixed at 230 Hz for tone A and 276 Hz for tone B. The pitch interval between these two tones was 316 cents, which can be easily judged to be higher or lower in pitch when they are presented as pure tones (*Ichimiya & Ichimiya, 2016*). The tones of the first four pairs (X series) were those from which partials with frequencies greater than 1,600 Hz were eliminated. From each of these four pairs, partials with frequencies less than 250, 500, 750, or 1,000 Hz were also eliminated. These were named X1, X2, X3, and X4, respectively. The tones of the latter four pairs (Y series) were the ones from which partials with frequencies greater than 1,300 Hz were eliminated. Partials with frequencies less than 250, 500, 750, or 1,000 Hz were also eliminated from each pair. These were named Y1, Y2, Y3, and Y4, respectively. These settings were aimed at constructing each tone pair such that the tone with the lower F0 (*i.e.,* tone A) was higher in terms of the frequency of the lowest partial. The frequencies and harmonic ranks of the partials are shown in Fig. 1. For
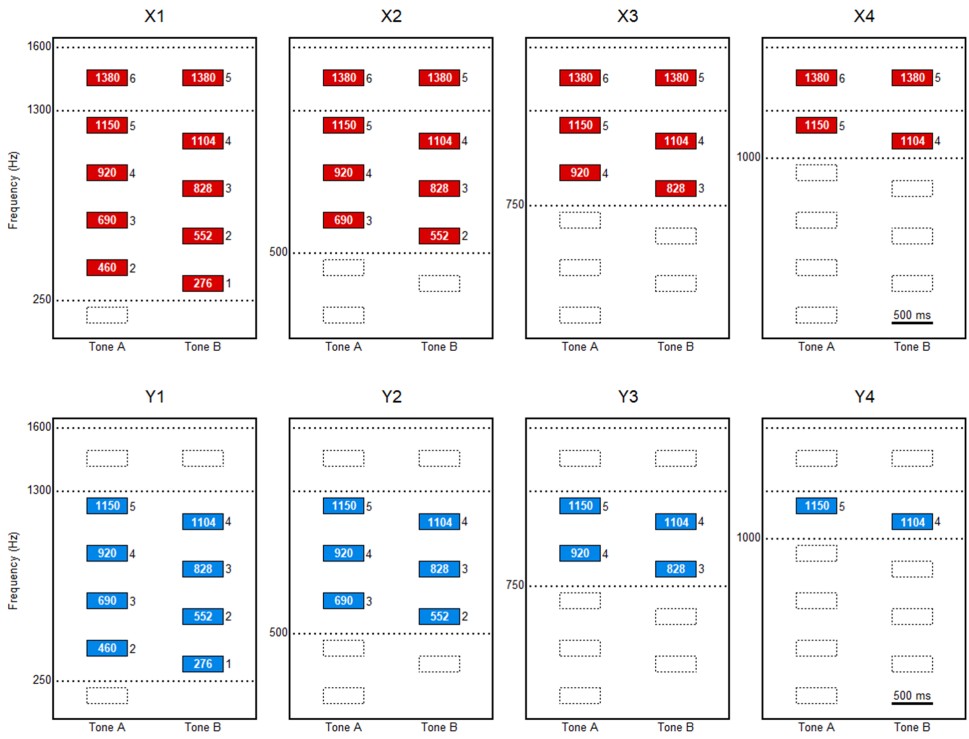

**Figure 1** **Schema of the tone pairs for experiment 1.** A total of eight tone pairs are illustrated. These are the tones from which the higher and lower partials are eliminated for the creation of the harmonic complex tones. The fundamental frequency (F0) of each tone pairs is the same; 230 Hz for tone A and 276 Hz for tone B. The tone pairs, X1, X2, X3, and X4, are the ones from which partials with frequencies greater than 1,600 Hz are eliminated. Partials with frequencies less than 250, 500, 750, or 1,000 Hz are also eliminated from X1, X2, X3, and X4, respectively. The tone pairs, Y1, Y2, Y3, and Y4, are the ones from which components with frequencies more than 1,300 Hz are eliminated. Partials with frequencies less than 250, 500, 750, or 1,000 Hz are also eliminated from Y1, Y2, Y3, and Y4, respectively. The numbers in the columns represent the frequencies of the partials, and the numbers to the right of the columns represent the harmonic ranks of the partials.

example, tone A consists of the partials of 1,150 and 1,380 Hz, which are the 5th and 6th harmonics of F0 for X4. The top frequency, which is the frequency of the highest partial, is the same as for the X series (1,380 Hz) but is higher in tone A than in tone B as for the Y series (1,150 Hz *vs.* 1,104 Hz). Only one pair, Y4, consists of pure tones, whereas the other pairs consist of complex tones.

For experiment 2, tonal sequences were constructed using tones of 500- or 1,000-ms duration, with a 10-ms rise and fall time. Thirteen tones were connected to create a passage of "Lightly Row". This passage was selected for the experiment because of the participants' familiarity with the song. The song, which is also known as "Butterfly", is a popular children's song in Japan. Similar to experiment 1, the higher and lower components of the harmonic complex tones were eliminated from the complex tones. A total of eight versions were presented in experiment 2. The versions were classified as X1, X2, X3, X4, Y1, Y2, Y3, and Y4 in the same manner as in experiment 1 (Fig. 2).

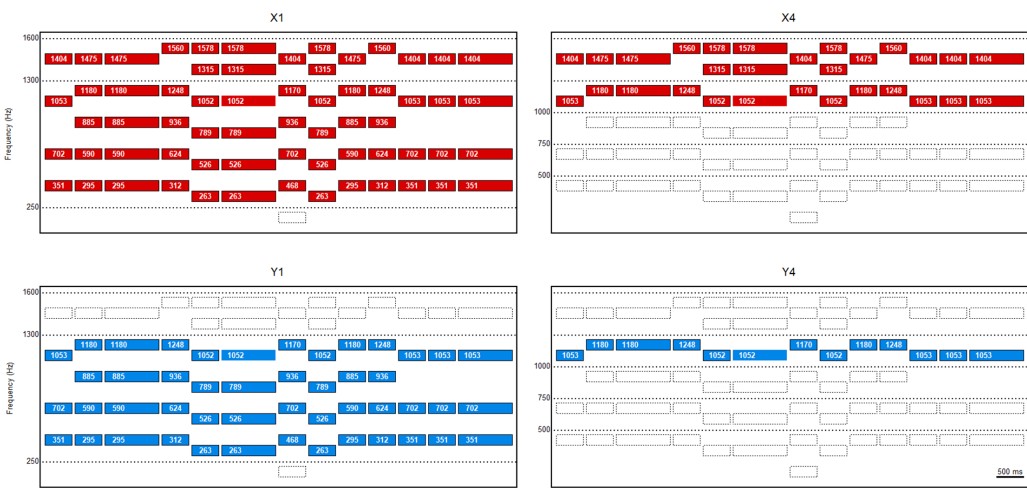

**Figure 2** **Schema of the connected tones for experiment 2.** Thirteen tones are connected to make a passage of "Lightly Row". Similar to experiment 1, the higher and lower partials of the harmonic complex tones are eliminated from the complex tones. Among the eight versions used in the experiment, versions X1, X4, Y1, and Y4 are illustrated.

An Aspire S3 computer (Acer America Corporation, San Jose, CA, USA) with a USB audio processor (SE-U55SXII; Onkyo Digital Solutions, Tokyo, Japan) was used to deliver auditory stimuli binaurally through dynamic headphones (MDR-7506; Sony, Tokyo, Japan) at a comfortable level for the participants, which was approximately 75 dB SPL in all cases.

## Experimental procedure

The experiments were performed as described in our previous studies (*Ichimiya & Ichimiya, 2019*; *Ichimiya & Ichimiya, 2023*).

In experiment 1, the computer monitor showed two buttons that played one of the eight pairs of prepared tones. The participants were asked to compare these tone pairs by clicking on the buttons and select the tone that was higher in pitch. The number of times the participant heard the tone was not predetermined, and the participants were allowed to click on the buttons multiple times before making a decision. Each task was performed twice for each of the eight pairs, resulting in a total of 16 tasks. The order of the tone pairs and the order of the two buttons were randomized.

In experiment 2, the computer monitor displayed a button that played one of the eight versions of "Lightly Row". The participants were asked if the passage they heard sounded correct based on the melody or off-key. The number of times of hearing the tone was not predetermined, and the participants were allowed to click on the buttons multiple times before making a decision. Each task was presented twice; thus, a total of 16 tasks were performed. The order of the tasks was randomized.

The participants received written instructions *via* a computer monitor during both experiments and were asked to respond to the questionnaire in writing.

## Statistical analysis

All statistical analyses were performed using EZR version 1.52 (Saitama Medical Center, Jichi Medical University, Saitama, Japan) (*Kanda, 2013*), a graphical user interface for R version 4.02 (*R Core Team, 2020*). Since the data obtained were sporadic, nonparametric tests were applied for statistics. Wilcoxon's signed rank test and McNemar's chi-square test were used for the statistical comparisons.

## RESULTS

In experiment 1, the perception was defined as a pitch shift when the participants selected tone A as the tone that is higher in pitch, as the tone they selected was lower in frequency in terms of F0. The participants' responses were considered to be spectral as they appeared to have responded based on the frequency of partials that make up the tones. Conversely, the perception was defined as an F0 response when they selected tone B as the tone that is higher in pitch. *Schneider et al. (2005)* added each participant's responses and computed an index that expresses the proportion of spectral and F0 responses on a scale ranging from $-1$ to $+1$. We referred to this score as the Shift Index (SI), which is identical to the Schneider Index described by *Ladd et al. (2013)*. The formula used is as follows:

$$SI = \frac{(sp - f0)}{(sp + f0)}$$

where sp is the number of spectral responses, and f0 is the number of F0 responses. Eight SI values were calculated (*i.e.,* values for X1, X2, X3, X4, Y1, Y2, Y3, and Y4) for each participant. These values could be $-1$, 0, or $+1$ since each task was performed twice.

The average SI values for all participants are shown in Fig. 3. In the X series, in which partials with frequencies greater than 1,600 Hz were eliminated, the SI values were higher when more partials were eliminated at low frequencies. For statistical analysis, Wilcoxon's signed rank test was applied for each of the two pairs using matched samples from the participants. Compared with those for X1, the SI values were significantly higher for X2 ($p = 0.037$), X3 ($p = 0.001$), and X4 ($p = 0.002$). The results were similar in the Y series, in which partials with frequencies greater than 1,300 Hz were eliminated. Compared with those for Y1, the SI values were significantly higher for Y3 ($p = 0.008$) and Y4 ($p < 0.001$). A comparison between the X and Y series was performed to analyze the effect of eliminating the partials with high frequencies. The SI values were significantly higher when Y1 was compared with X1 ($p = 0.005$), Y2 was compared with X2 ($p = 0.023$), Y3 was compared with X3 ($p = 0.023$), and Y4 was compared with X4 ($p < 0.001$).

An ambiguous response was defined as a difference in the participant's responses to the same task, which had been performed twice. The percentages of these ambiguous responses, with the percentages of unambiguous F0 and spectral responses for each tone pair, are shown in Fig. 4. The data were compared with those of tone pair Y4, which was supposed to be the tone pair with the least ambiguous response as it consisted of pure tones. McNemar's test was applied to analyze the paired nominal data (*i.e.,* ambiguous or unambiguous) from the participants. The ambiguous response was significantly higher for X3 ($p = 0.027$) and X4 ($p = 0.027$).

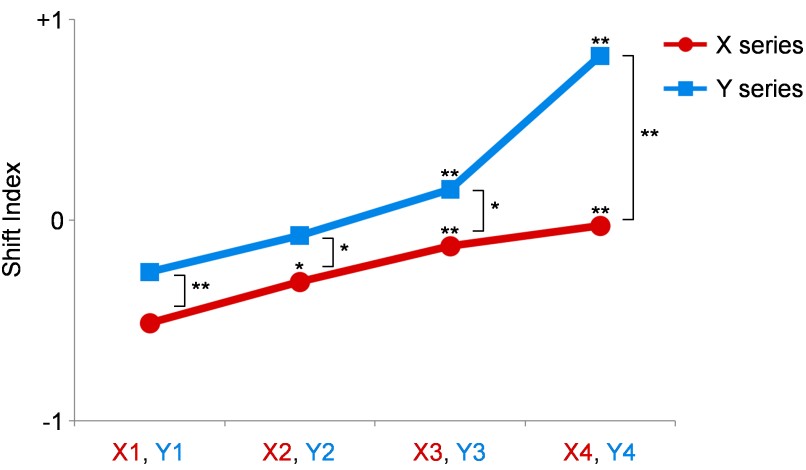

**Figure 3  Results of experiment 1.** The shift index (SI) values of the X series are shown as red circles, and those of Y series are shown as blue squares. In the X series, the SI values are higher when more partials are eliminated at low frequencies. Compared with those for X1, the SI values are significantly higher for X2, X3, and X4. Similarly, compared with those for Y1, the SI values are significantly higher for Y3 and Y4. The SI values are significantly higher when X1 is compared with Y1, X2 with Y2, X3 with Y3, and X4 with Y4. *, $p < 0.05$; **, $p < 0.01$ (Please refer to the text for the exact $p$-values).

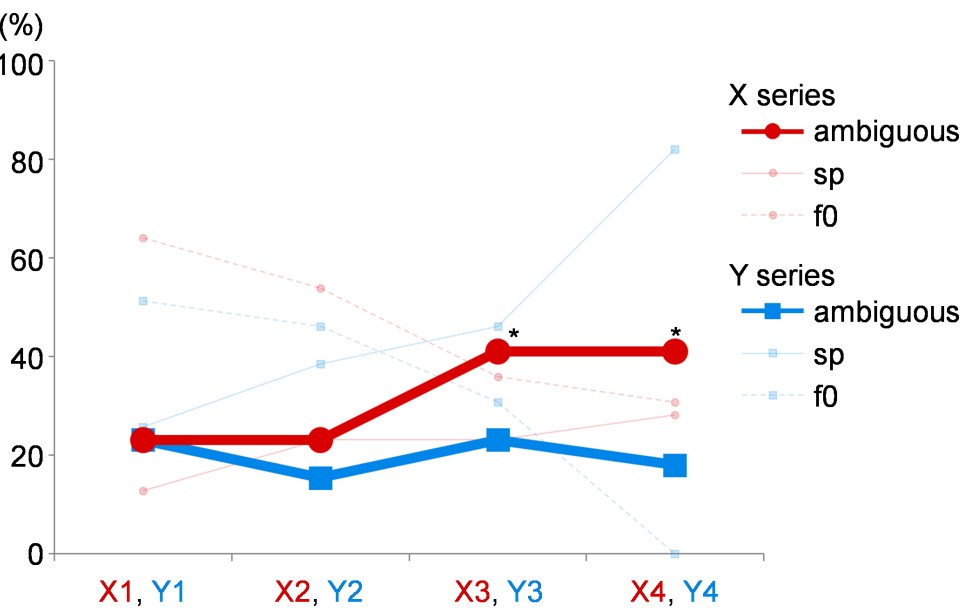

**Figure 4  The ambiguous responses in experiment 1.** When the responses to the same task, which is repeated twice, differ, it is defined as an ambiguous response. The percentages of these ambiguous responses are shown. The values of X series are shown as red circles, and those of Y series are shown as blue squares. The percentages of unambiguous F0 and spectral responses for each tone pair are shown as transparent lines for reference (f0 and sp). Compared with the results of tone pair Y4, the ambiguous response is significantly higher for the tone pairs X3 and X4. *, $p = 0.027$.

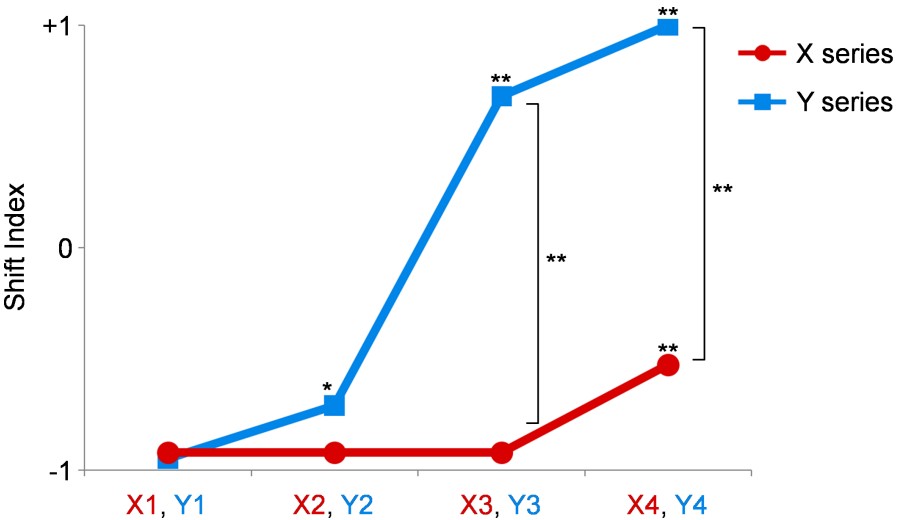

**Figure 5** **Results of experiment 2.** The shift index (SI) values of the X series are shown as red circles, and those of Y series are shown as blue squares. The SI values are significantly higher for X4 compared with those for X1. The SI values are also significantly higher for Y2, Y3, and Y4 compared with those for Y1. The SI values are significantly higher when X3 is compared with Y3 and X4 with Y4. *, $p = 0.025$; **, $p < 0.01$ (Please refer to the text for the exact $p$-values).

In experiment 2, the perception was considered as a pitch shift caused by spectral response when the participants judged that the melody sounded off-key. They were analyzed similarly as in experiment 1. The calculated SI values are shown in Fig. 5, and the results were similar to those of experiment 1. The SI values were significantly higher for X4 than those for X1 ($p = 0.003$). The SI values were significantly higher for Y2 ($p = 0.025$), Y3 ($p < 0.001$), and Y4 ($p < 0.001$) than those for Y1. Moreover, the SI values were significantly higher when Y3 was compared with X3 ($p < 0.001$) and Y4 was compared with X4 ($p < 0.001$). The participants' ambiguous responses were analyzed similarly as in experiment 1, as shown in Fig. 6. Compared with the results of Y4, they were significantly higher for X4 ($p = 0.002$) and Y2 ($p = 0.020$).

To examine the effect of age on the study outcomes, the data from experiments 1 and 2 were re-analyzed with the participants divided into two age groups: $\leq 39$ years and $\geq 40$ years.

No evident differences were observed between the two age groups for any of the experiments. The SI values that showed large differences when all participants were grouped together regardless of age also showed significant differences within both age groups. The $p$-values obtained using Wilcoxon's signed rank test are shown in the order of $\leq 39$ years group and $\geq 40$ years group. In experiment 1, the SI values were significantly higher for Y4 than those for Y1 ($p = 0.001$, $p = 0.002$). The SI values were significantly higher when Y4 was compared with X4 ($p = 0.001$, $p = 0.009$). In experiment 2, the SI values were significantly higher for X4 than those for X1 ($p = 0.032$, $p = 0.048$). The SI values were significantly higher for Y3 ($p < 0.001$, $p < 0.001$) and Y4 ($p < 0.001$, $p < 0.001$) than those for Y1. The SI values were significantly higher when Y3 was compared with X3

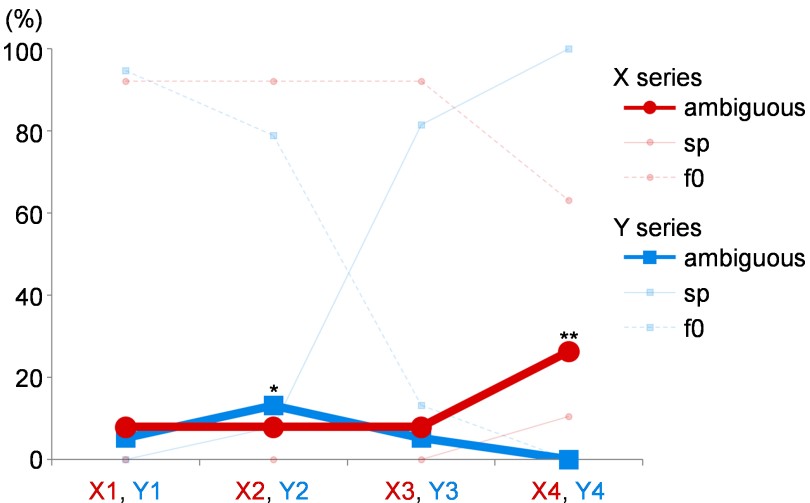

**Figure 6** **The ambiguous responses in experiment 2.** The percentages of ambiguous responses are shown. The values of the X series are shown as red circles, and those of Y series are shown as blue squares. The percentages of unambiguous F0 and spectral responses for each tone pair are shown as transparent lines for reference (f0 and sp). Compared with the results of version Y4, the ambiguous responses are significantly higher in ratio in the cases of X4 and Y2. *, $p = 0.020$; **, $p = 0.002$.

($p < 0.001$, $p < 0.001$) and when Y4 was compared with X4 ($p < 0.001$, $p < 0.001$). No significant differences were observed when the ambiguous responses were analyzed.

## DISCUSSION

This study demonstrated that pitch shift can be provoked by complex tone stimulation *via* the removal of partials. Taking into consideration that these experiments enrolled healthy participants, these results suggest that pitch shifts can be perceived when patients with hearing loss hear certain complex tones, regardless of the underlying etiology of the hearing loss. To examine the effect of age on the study outcomes, the data were re-analyzed with the participants divided into two age groups. Although this broad grouping may not capture the detailed age effect, we observed no evident differences between the two age groups. Thus, it is unlikely that age would have a substantial impact on the results of the present study. Low-tone hearing loss of various severities was simulated, which prevented the participants from hearing the lower components of complex tones. The elimination of components with frequencies of less than 250 Hz simulated mild low-tone hearing loss, whereas the elimination of components with frequencies of less than 500, 750, or 1,000 Hz simulated more severe low-tone hearing loss as the values increased. The elimination of high-frequency components can be considered as the simulation of high-tone hearing loss. In addition, it can also be considered to simulate modified sounds that patients may hear in everyday life in certain environments. Sounds are subject to reflections and refractions caused by walls or objects in their paths. Thus, the sound "image" that reaches the ear will differ somewhat from that initially generated. Diffraction occurs at lower frequencies because lower-frequency sounds have longer wavelengths (*Moore, 2012*). Thus, objects in

the path of sound may act as low-pass filters after the bending of sound around them. Consequently, high-frequency tones are eliminated. In this study, we used test tones in which partials with a frequency greater than 1,300 Hz and 1,600 Hz were eliminated. The former simulates sounds that reach the ear through thicker walls or objects in their path than the latter.

The results of experiment 1 suggest that when the simulated low-tone hearing loss is mild, many participants do not perceive pitch shift as they perceive the missing fundamental tone. However, when the simulated low-tone hearing loss is more severe, many participants perceive pitch shifts as they cannot perceive the missing fundamental tone. These results were more apparent in the case of the Y series (*i.e.,* the participants heard the tones from which partials with a frequency greater than 1,300 Hz were eliminated).

Individual differences have been suggested for the perception of auditory stimuli that lack F0. Some individuals readily identify the pitch of such tones with the missing F0 (F0 listeners), whereas some individuals base their judgment on the frequency of the partials that make up the tones (spectral listeners) (*Schneider et al., 2005*). However, recent research has shown that classifying individuals as "F0 listeners" or "spectral listeners" is an oversimplification. In a study by *Ladd et al. (2013)*, the participants were asked to judge the pitch change in stimuli comprising two missing fundamental tones, which was constructed to reveal whether the pitch perception was based on missing fundamental or partials. They used missing fundamental tones of various top frequencies and confirmed that there are robust individual differences in the perception of missing fundamental stimuli; however, the participants gave predominantly spectral responses at lower top frequency levels and F0 responses at higher top frequency levels. Thus, it was concluded that two modes of perception ("F0 listening" and "spectral listening") may exist, both of which are available to many listeners. Similarly, in our study, many listeners also perceived both modes of perception according to the stimuli. The results presented here (*i.e.,* the pitch shifts when a low tone is not heard, especially in cases where a high tone is not heard) are reasonable because missing fundamental pitches are generally less distinct when fewer numbers of harmonics are present (*Hartmann, 1993*; *Schneider et al., 2005*; *Moore, 2012*). It may also be speculated that in our experimental protocol, the top frequency of the complex tones affects the pitch shift since the top frequency of tone A was always higher than that of tone B in the Y series, whereas the top frequencies of the complex tones were always the same in the X series. Therefore, further studies must be conducted in the future to evaluate the effect of controlling for the number of harmonics and the top frequency.

Ambiguous responses were analyzed using the data from our experiment. In the present study, we provisionally defined ambiguous responses as different responses to the same task. Responses were high for both X3 and X4. Although this was a preliminary analysis, based on our results, it can be inferred that the SI values for X3 and X4 were close to 0 as each participant's response was ambiguous in judging these tasks, rather than dichotomizing the participants' responses between F0 and spectral responses.

Our previous study (*Ichimiya & Ichimiya, 2019*) presented participants with harmonic complex tones that lacked low-tone components. Their perception of these tones revealed a pitch shift compared with the tone that was expected. It was observed that when these

tones were presented binaurally, with the low-tone components eliminated in one ear, approximately half of the participants reported hearing the tones at different pitches in both ears. Based on these findings, we hypothesized that, under specific circumstances, the stimulation of complex tones might lead to binaural diplacusis due to the pitch-shifted tones in one ear. Building upon our previous study, we conducted further investigations on pitch shift in the present study. Compared with the number of components in the present study, the number of components in the complex tones was smaller in our previous study, suggesting that many participants might not have perceived F0 when the low-tone components were eliminated in the previous study. The results of the present study emphasize the importance of considering the missing fundamentals when perceiving complex tones and indicate that complex tone stimulation can induce pitch shifts or binaural diplacusis with low-tone hearing loss under limited conditions in which the number of components is small.

Experiment 2 aimed to demonstrate the effect of the pitch shift observed in experiment 1 on everyday life. The results were similar to those of experiment 1 in terms of statistical significance. However, experiments 1 and 2 appear to exhibit pronounced differences. In experiment 2, the SI values indicate more marked F0 responses except for Y3 and Y4. The participants were asked whether the connected tones sounded correct based on the melody or off-key in experiment 2. Such criteria for overall judging might have increased the tendency toward F0 responses. The more unambiguous answers of experiment 2 may also be related to such judging criteria. The abrupt shift towards positive SI values in Y3 and Y4 may be related to the feature of the tonal sequences. Since pure tones were included in Y3 and Y4, it is possible that they may sound off-key. Thus, it is difficult to interpret the results of experiment 2 alone, but combined with the results of experiment 1, a possible scenario is illustrated in Fig. 7A. Patients with low-tone hearing loss do not perceive complex tones as off-key when they hear them from near a sound source as they perceive the missing fundamental. However, when the same patients hear the same tones away from the sound source through walls or objects that eliminate the high-frequency components from complex tones, they may perceive the tones as off-key.

Interestingly, the results of the present study can also lead to a completely different interpretation. In the case of some patients with cochlear disorders of low-tone hearing impairment with pitch shift, pure tones may be perceived as off-key, whereas complex tones may not. This phenomenon may be attributed to the fact that these patients are unable to hear low-tone components that should be shifted in pitch, but are able to hear upper harmonics of the complex tones that are not impaired (Fig. 7B).

Since the conditions that construct the tones under which the pitches shift were extreme in this study, they may not be frequently encountered by real patients. We simulated low-tone hearing loss in participants, preventing them from hearing the lower components of complex tones. However, real patients may hear lower components to some extent, unless their hearing loss is severe. Moreover, patients may not remain in an environment where high-frequency components have been eliminated for long periods of time. Thus, the pitch shift we have demonstrated here should only occur occasionally. Clinically, we sometimes see patients who complain of occasional pitch shift perception. This may
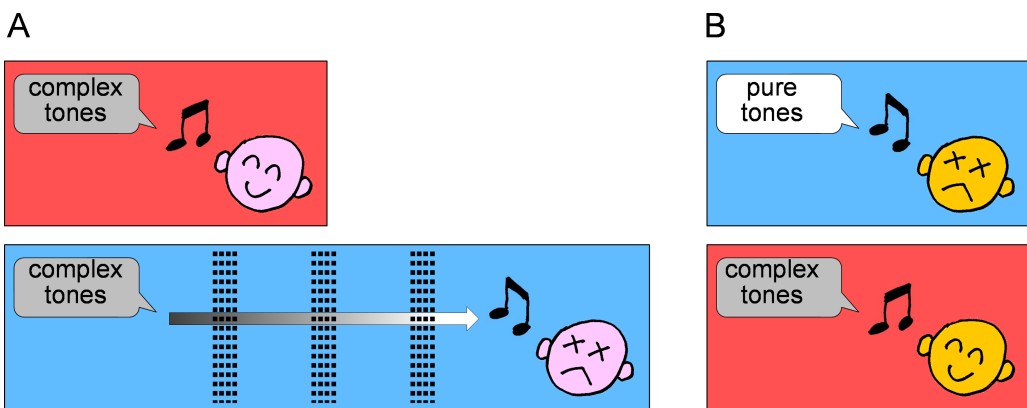

**Figure 7** **Possible pitch shift on hearing complex tones.** (A) There may be patients with low-tone hearing loss who do not perceive the tones as off-key when they listen to complex tones from near a sound source as they perceive the missing fundamental (upper illustration). However, when the same patients hear the same tones away from the sound source, through walls or objects which eliminate the high frequency components from complex tones, they may perceive the tones as off-key (lower illustration). (B) In the case of some patients with cochlear disorders of low-tone hearing impairment with pitch shift, pure tones may be perceived as off-key, whereas complex tones may not (lower illustration). This phenomenon may be attributed to the fact that these patients are unable to hear low-tone components that should be shifted in pitch, but are able to hear upper harmonics of the complex tones that are not impaired.

represent the pitch fluctuation in patients with Meniere's disease described by *Brännström & Grenner (2008)*, but it may be merely due to the change in the sound environment they are in, not due to the actual fluctuation in pitch. Detailed interviews must be conducted with these patients to elucidate the environment where they have perceived pitch shift. Evaluating pitch shift by hearing loss becomes extremely complicated when complex tones are considered stimulation tones. Thus, further research is needed to address this issue.

## CONCLUSIONS

We investigated the pitch shift provoked by complex tone stimulation in healthy participants by simulating low-tone hearing loss. We found that the pitch of complex tones shifts when a low tone is not heard, especially in cases when a high tone is also not heard. It is worth noting that the conditions of the tones under which the pitches shifted in this study were extreme and may not be encountered frequently by patients in a real-world setting. However, it may partly explain the pitch shift observed in patients with hearing loss. Since our experiments were performed on healthy participants, we can infer that such pitch shifts can be perceived regardless of the underlying etiology of the hearing loss. Thus, pitch shifts associated with hearing loss should be interpreted with caution.

## ACKNOWLEDGEMENTS

We would like to thank Editage for English language editing.

### Funding

The authors received no funding for this work.

### Competing Interests

Issei Ichimiya and Hiroko Ichimiya are employed by the Ichimiya Clinic. The authors declare there are no competing interests.

### Author Contributions

- Issei Ichimiya conceived and designed the experiments, performed the experiments, analyzed the data, prepared figures and/or tables, authored or reviewed drafts of the article, and approved the final draft.
- Hiroko Ichimiya performed the experiments, analyzed the data, prepared figures and/or tables, and approved the final draft.

### Human Ethics

The following information was supplied relating to ethical approvals (*i.e.*, approving body and any reference numbers):

The study protocols were reviewed and approved by the Clinical Research Ethics Committee of Ichimiya Clinic (H3001-1).

### Data Availability

The raw data are available in the Supplemental Files.

### Supplemental Information

Supplemental information for this article can be found online at http://dx.doi.org/10.7717/peerj.16053#supplemental-information.

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
