# Peer review of "Simulation of hearing loss can induce pitch shifts for complex tones"

_PeerJ, doi:10.7717/peerj.16053_

## Round 0.1 · original submission · Major Revisions

· Academic Editor

Major Revisions

Dear authors, I apologize for the delay in completing the review to your work. Unfortunately in the course of the review one reviewer did not complete the review and a second withdrew availability to review after initially agreeing.

Now both contacted reviewers have completed their review. Please find their comments below.

Best regards,
Emiliano Brunamonti

Reviewer 1 ·

Basic reporting

The authors report the results of a behavioral study in which subjects were asked to indicate which of two complex tones was higher in pitch. Different higher and lower partials were removed from tones pairs to simulate hearing loss, and the results showed that pitch shift is perceived when more severe low tone hearing loss is simulated, as well as when higher tones are not heard.
The manuscript is generally well written and I believe the Introduction does a good job of providing the appropriate context for the article, at least for a general reader who is not an expert in the field. I nevertheless have some concerns about the experimental design, results, and discussion sections (which will be better explained in later sections).
Raw data are provided, are easily understandable, and allow one to check the main results presented in the figures (I have noticed, however, that in the raw data of exp1 the results of subjects 38 and 39 seem to be duplicated. The authors may want to fix the file by removing the last two rows).
I have two major concerns about figures:
1 In Fig 4 and 6 the proportion of ambiguous responses are reported, along with other lines that are not clear what they represent (transparent sp and f0 for the x and y series). I guess they represent the proportion of non-ambiguous sp and f0 responses for each tone pair? If so, this should be clearly described in the figure legend and in the text where the figure is referred to.
2 Figure 7 and 8 do not appear to provide any useful additional information to the concepts already expressed in the text; actually, fig7 fails to provide a graphic illustration of a low pass band filter: complex tones through the walls lose high frequencies, but the depicted arrow goes through the walls without undergoing any change whatsoever. The authors should consider whether to redo the figures (and perhaps combine them into a single summary figure) or eliminate them.

Experimental design

Several aspects of the experimental design are not clear enough to me. I will list them in order of importance.

1 The authors calculated the SI index in different experimental conditions (X1-X4 and Y1-Y4) and applied the Wilcoxon signed rank test to determine any significant differences. It seems to me that the value of SI was not calculated for each individual subject and then mediated, but rather by collapsing the responses of all subjects together to obtain a single value per experimental condition (this is the reason why Fig3 and Fig5 do not show any error bars on the values?). I do not understand how this type of test can be used to compare two individual values, given that Wilcoxon is conducted on repeated measures. Alternatively, the values shown in Fig3 and Fig5 are mean values between individual subjects, each with its own specific SI index (which, however, can assume only 3 values, -1,0 or 1 since each pair of tones is tested only twice). In the latter case, the authors might explain why they chose to perform only two repetitions for each subject for each pair of tones. A greater number of repetitions would have enabled us to estimate a more accurate SI index for each subject.
In any case, I think that in order to make everything clearer for the reader, the authors should substantially expand the section “statistical analysis” (only two rows [158-160] are currently explaining the statistical tests adopted) and the beginning part of the results section, where it is explained how the SI index is calculated (particularly in line 167 “...added each participant’s answers...”: do you mean the two answers of each subject or the responses of all subjects together?)

2 A second important point is the difference between the two experiments. The authors should better explain the scientific rationale and the experimental questions behind the choice to conduct two experiments and how they differ from each other. It is not clear to me what the results of exp2 add to the results of exp1. In the Discussion, only a small part is dedicated to commenting on the results of exp2 [rows 262-268], in which it is essentially said that the results are not different from those of exp1. Despite that, the results in Fig3 and Fig5 seem to differ in several aspects (for example, the SI index values in the Y series of exp2 shift abruptly towards positive values in Y3-4 ; or the SI values of the X series indicate a much more marked f0 response in exp2 than exp1). Can the authors comment on these differences?

3 Regarding the presentation of stimuli, subjects were allowed to click the button to listen back to the pairs of tones many times, but how many? Could each subject decide freely, without a maximum number or a time limit? Not having this experimental variable under control could create various biases, especially when the authors classify the participant’s responses as ambiguous or not. Subjects who listened several times to both tones could show more congruent responses between the two repetitions than subjects that did it less often, and the same bias could be reflected between the different experimental conditions. Is it possible to report the information regarding the number of total listens for each subject and/or experimental condition?

4 The age control needs to be thoroughly explored. Dividing subjects in younger or older than 40 does not give any information about the real age gap (mean age of 38 vs 41 is different from mean age of 30 vs 50): authors should provide the mean and standard deviation of the two age groups. In the result section [rows 199-200] it is just stated that there are no significant differences with a fisher exact test, which is different from the Wilcoxon adopted for the main results. If the same analysis were done, only divided into two subgroups, why a different test?

Validity of the findings

Conclusions are well stated; however, I believe that the Discussion could be improved by better explaining the novelty and impact of the result of this study, especially compared with the previous literature of the authors. In the Discussion [rows 253-261] it is stated that this study expand the result of the previous one, but it seems that the discussion revolves around the results of the 2019 study. From the text: “The results of the present study suggest that complex tone stimulation can induce binaural diplacusis with low-tone hearing loss under limited conditions in which the number of components is small”. Binaural diplacusis was not tested in the current study. If the authors believe that the result of the current study may be useful for a reinterpretation of the 2019 paper, this part needs to be revised, pointing out the differences between the two studies.

Reviewer 2 ·

Basic reporting

The reporting is generally satisfactory.
- I would suggest changing the title to "Simulation of hearing loss can induce pitch shifts for complex tones". Also, in Discussion and elsewhere, I would not use the term "hearing is disturbed". This is ambiguous and implies that someone is perturbing cochlear function, for instance. Instead, be direct and say "hearing loss is simulated" or "partials are removed".
- Also, in the introduction, please define the term "pitch shift". Do you mean pitch shift over time, pitch shift from what is expected, or pitch shift between ears? Similarly in the abstract.
- While the acronym MF is defined, it is not a common one. I would suggest just writing this out as missing fundamental.

Experimental design

The research question is defined sufficiently.
Some issues in methods:
- Age range and standard deviation should be included. In addition, as older subjects were tested, this brings up the question of whether all subjects had normal hearing. Were listeners verified to have normal hearing using pure tone audiometry? Was tympanometry conducted to confirm normal function of the tympanic membrane? Abnormal tympanometry can lead to diplacusis.
- It is not clear what sequence the various tones were presented (randomized?), how many times/repetitions. These details need to be added.

Validity of the findings

- In the Discussion and elsewhere, I find it strange that the authors refer only to low-frequency hearing loss when the majority of hearing loss is high-frequency loss. So it would make sense in lines 203-222 to discuss the removal of the higher frequency partials as also simulating the more common type of hearing loss, before discussing sound attenuation through walls. The latter can be brought up, but the high-frequency hearing loss should be brought up first as the more likely scenario. This change in emphasis needs to be applied throughout the Discussion.
- Future studies should consider controlling for the number of harmonics and the top frequency, as mentioned in lines 242-245.

Additional comments

Discussion is a bit long. I would condense and reduce speculation in lines 269-312.

---

## Round 0.2 · Minor Revisions

· Academic Editor

Minor Revisions

Dear Dr. Ichimiya,

There are a few small requests from a reviewer before accepting your manuscript for publication.

Please prepare an updated version of the manuscript with the required changes.

Best regards,
Emiliano Brunamonti

Reviewer 1 ·

Basic reporting

The manuscript is improved after the revision based on previous suggestions and comments. I believe that some minor corrections are still needed:
• The mean and std of the two age groups is still missing. Please add it in the materials and methods section (row 94)
• This sentence is not clear in the discussion (row 227): "Although this broader grouping may not capture the detailed age effect, we observed no evident differences between the two age groups."
Why broader grouping? One broad group (all subjects) was divided into two more narrow groups, in order to capture the age effect.
• "Thus, it is unlikely that the age effect would have a substantial impact on the results of the present study" (row 227): remove the word "effect" after age.
• Reformulate row 210 for more clarity, suggestion: “The SI values that showed large differences when all subjects were grouped together regardless of age also showed significant differences within both age groups”
• The authors claim that the main difference between the two experiments is that Experiment 2 aimed to demonstrate how the pitch shift observed in experiment 1 could affect everyday life: this is ok but it should be stated clearly earlier in the text since it is currently only explained in the discussion part (maybe in the material and methods section, or a the end of the introduction)

Experimental design

No comment

Validity of the findings

No comment

Additional comments

No comment

---

## Round 0.3 · accepted · Accept

· Academic Editor

Accept

The authors have responded to all reviewer comments and believe that the current version of the manuscript may be suitable for publication.
Congratulations,
Emiliano Brunamonti